# Pipeline Combinators for Gradual AutoML

**Guillaume Baudart**
Inria, ENS – PSL University, France
guillaume.baudart@inria.fr

**Martin Hirzel**
IBM Research, USA
hirzel@us.ibm.com

**Kiran Kate**
IBM Research, USA
kakate@us.ibm.com

**Parikshit Ram**
IBM Research, USA
parikshit.ram@ibm.com

**Avraham Shinnar**
IBM Research, USA
shinnar@us.ibm.com

**Jason Tsay**
IBM Research, USA
jason.tsay@ibm.com

## Abstract

Automated machine learning (AutoML) can make data scientists more productive. But if machine learning is totally automated, that leaves no room for data scientists to apply their intuition. Hence, data scientists often prefer not total but gradual automation, where they control certain choices and AutoML explores the rest. Unfortunately, gradual AutoML is cumbersome with state-of-the-art tools, requiring large non-compositional code changes. More concise compositional code can be achieved with combinators, a powerful concept from functional programming. This paper introduces a small set of orthogonal combinators for composing machine-learning operators into pipelines. It describes a translation scheme from pipelines and associated hyperparameter schemas to search spaces for AutoML optimizers. On that foundation, this paper presents Lale, an open-source sklearn-compatible AutoML library, and evaluates it with a user study.

## 1   Introduction

Automated machine learning can make data scientists more productive by lowering the barrier to entry for novices as well as reducing the tedium for experts. With *total* automation, the data scientist just provides data and the AutoML tool does the rest. While this yields a convenient starting point, it gives users little control. Instead, *gradual* automation, where the user tweaks some choices by hand, leverages user expertise for better outcomes and wastes less compute time. Unfortunately, gradual automation is not easy with existing AutoML tools and their programming interfaces. Existing AutoML tools tend to have different interfaces from those for manual machine learning, preventing a gradual transition between the two. Furthermore, with existing tools, code changes for gradual automation tend to be large and non-compositional. The programming interface of existing tools is often coupled to a given optimizer. And finally, tools often support controlling some aspects of automated pipeline search only with a much lower-level programming interface, if at all.

We hypothesize that gradual AutoML can be made easier via combinators. Combinators were first discovered by Schönfinkel in the 1920s in an attempt to find elementary building blocks of logic [41]. Combinators are functions that compose functions and play an important role in functional programming. For instance, Lisp provides the combinators `map` and `reduce` [43], which inspired the MapReduce programming model [11, 22]. Combinators enable a *tacit* programming style, where data remains unnamed, keeping function compositions concise and minimizing edits when re-arranging them. We argue that sklearn's `make_pipeline` and `make_union` [7] are combinators: they compose machine-learning operators into pipelines without naming datasets such as `X` or `y` during composition. These two combinators enable the two primary execution modes, training and prediction, of sklearn pipelines. This paper streamlines them with infix notation, adds a choice combinator, and adds AutoML search as a third execution mode.

35th Conference on Neural Information Processing Systems (NeurIPS 2021).

Enabling search as a third execution mode requires a translation from combinators to AutoML search spaces. To solve this challenge, this paper describes a novel translation scheme, which takes a combinator-based pipeline and generates a search space suitable for a given AutoML optimizer. The translation first normalizes and combines the hyperparameter *schemas* of the operators in the pipeline, which specify the space of hyperparameter values. Next, depending on the optimizer at hand, it flattens and discretizes the search space. This translation is essential to bridge from the modular and tacit combinators syntax to an optimizer's syntax, and may re-arrange components and introduce (possibly mangled) names as needed. The translation scheme also supports round-tripping back to combinators, needed for evaluating and understanding results as well as for iterative refinement.

This paper introduces Lale, a Python AutoML library implementing the combinators and the translation scheme. Lale enjoys active use both in the open-source community (https://github.com/ibm/lale/) and in IBM's AutoAI product. It is sklearn-compatible, giving users access to many operators and familiar functionality. Lale is gradual, letting users specify only what they want while reusing and automating the rest. For instance, Lale comes with an extensive library of reusable hyperparameter schemas for many popular operators, so users rarely need to write their own schemas; but it also makes it easy to customize schemas when desired. There are Lale optimizer backends for Hyperopt [6] (with 4 solvers tpe, atpe, rand, and anneal); sklearn's GridSearchCV and HalvingGridSearchCV [7]; ADMM [26] (with 5 continuous and 5 combinatorial solvers); SMAC [15]; and Hyberband [25].

The contributions of this paper are:

- Combinators for gradual automated machine learning.
- A translation scheme for mapping combinators to a broad variety of optimizers.
- The Lale library and a user study for validating combinators and gradual automation.

This paper addresses the problem of simplifying gradual AutoML. Our solution, Lale, is easy to use for data scientists with moderate sklearn experience, as demonstrated by the user study. Lale uses combinators for modularity, and uses schemas both for search spaces and for type-checking. It is more expressive than the high-level interfaces of prior AutoML tools. Lale gives users fine-grained control over AutoML without requiring them to be AutoML experts.

## 2    Pipeline Combinators

This paper proposes three combinators to compose machine-learning operators into pipelines. To support hyperparameter tuning, each operator has an associated hyperparameter schema. Thus, when combinators compose operators, then under the hood, that implies weaving together the associated hyperparameter schemas. This section describes the syntax both for combinators and for schemas, leaving it to Section 4 to describe the translation that weaves them together for AutoML.

**Combinators.**    The three combinators are `>>`, `&`, and `|`. The *pipe combinator*, `p >> q`, connects the output of p to the input of q, similar to sklearn's `make_pipeline`. The *and combinator*, `p & q`, composes p and q side-by-side without creating dataflow between them. It is similar to sklearn's `make_union`, except that `make_union(p, q)` implicitly concatenates the output features of p and q whereas `p & q` does not. Instead, if users want to combine features after `&`, they have to explicitly add an operator for that. Making this explicit makes code more understandable and also provides flexibility to pick different data-combining operators when desired, for instance, vertical stacking vs. relational joining [40]. The *or combinator*, `p | q`, introduces a choice between p and q. While it has no direct equivalent in sklearn, it can be encoded in sklearn's `GridSearchCV`, as well as in other optimizers such as Hyperopt.

Figure 1 shows the syntax for combinator-based pipelines. The start symbol, *pipeline*, can be an individual operator or it can nest other pipelines by using one of the three combinators. An *individualOp* has a name and an optional configuration for zero or more hyperparameters. For example, in Figure 2, the operators `Project` and `AdaBoostClassifier` are explicitly configured with hyperparameters, whereas the operators `PCA`, `NoOp`, etc. are not. When the user does not manually configure hyperparameters, then AutoML automatically tunes them. A *hyperParam* configuration uses an *expression*, which can be a simple Python value or a nested pipeline. For example, in Figure 2, `Project` is configured with a simple Python dictionary, whereas `AdaBoostClassifier` is configured with a nested pipeline. An operator that takes another operator as an argument is *higher-order*, and besides `AdaBoostClassifier`,

| | |
|---|---|
| *pipeline* ::= *individualOp* \| *pipe* \| *and* \| *or* | |

```
pipeline     ::= individualOp | pipe | and | or
individualOp ::= NAME | NAME ( hyperParam* )
hyperParam   ::= NAME = expression
expression   ::= VALUE | pipeline
pipe         ::= pipeline >> pipeline
and          ::= pipeline &  pipeline
or           ::= pipeline |  pipeline
```

Figure 1: Pipeline syntax. Non-terminals are in lower-case *italics*, literal tokens (`(`, `)`, `=`, `>>`, `&`, `|`) are in `type-writer` font, other terminals are in all-caps *ITALICS*, and the meta-syntax (`::=`, `|`, *) indicates rules, choice, and Kleene star.

```
1  pre_n = (Project(columns={"type":"number"})
2          >> (PCA | NoOp))
3  pre_s = (Project(columns={"type":"string"})
4          >> (OrdinalEncoder|OneHotEncoder))
5  pplan = ((pre_n & pre_s)
6          >> ConcatFeatures
7          >> AdaBoostClassifier(
8              base_estimator=(J48 | LR)))
```

Figure 2: Pipeline example with two preprocessing sub-pipelines. Features from both are concatenated and piped to a boosted ensemble, whose base estimator is a choice of classifiers.

there are plenty of other examples, such as sklearn's `RFE`, `OneVsRestClassifier`, or `ColumnTransformer`, or fairness mitigation operators with preprocessing operators as an argument [19].

What makes the combinator-based pipeline syntax powerful is that everything nests. A pipeline is itself an operator in the sense that it implements the methods `fit` and `transform` or `predict`. Unlike other AutoML tools, Lale can thus easily express search spaces around higher-order operators. The syntax is concise, as it focuses just on operators and is tacit about data. The Lale library implements this syntax as pure Python code. That has the advantage that users need not learn a new programming language, and can instead make full use of Python's existing features. For example, Figure 2 uses a Python assignment (`pre_n = ...`) to improve readability. Furthermore, existing Python tooling just works, such as auto-completing editors, syntax and type checkers, and interactive notebooks.

**Execution modes.** Pipelines have three execution modes: predict or transform; fit; and AutoML search. The first two modes come from sklearn. Their semantics are based on the dataflow graph of a pipeline, which is a directed acyclic graph whose vertices are operators and whose edges are given by the `>>` combinator (note that `&` and `|` do not induce dataflow edges, and pipelines with `|` only support the third execution mode). The first execution mode of a pipeline, predict or transform, simply invokes predict or transform on the vertices in topological order, guaranteeing that the results from all predecessors of an operator are available before that operator fires. The second execution mode, fit, also processes vertices in topological order, calling first fit and then transform or predict. The result of fit on a pipeline is a new pipeline that is the same as the original except that all operators are replaced by trained versions, i.e., their learned coefficients are bound. Higher-order operators have the flexibility to invoke the methods of nested operators as needed for the outer operator.

The third execution mode of a pipeline, AutoML search, is implemented by translating the planned pipeline to a search space for an AutoML optimizer. The search space encodes joint operator selection and hyperparameter tuning. Given a planned pipeline such as `pplan` from Figure 2, users access this execution mode by invoking the `auto_configure` method:

```
1  best_found = pplan.auto_configure(
2      train_X, train_y, optimizer=Hyperopt, cv=3, max_opt_time=300, max_eval_time=30)
```

The result (in this case `best_found`) is the trained pipeline that yielded the best score (here, cross-validated with `cv=3`) during the search. Section 4 discusses the details of the translation.

**Schemas.** Besides combinators, which specify a space of pipelines to search, AutoML also needs hyperparameter schemas, which specify a space of hyperparameter values to search. Here, we adopt the *value-set* definition of a schema (or type) as a space (or set) of values [33]. While data scientists frequently write code with combinators to exercise control, they write schemas less often. As of today, Lale includes schemas for 216 operators, sufficient for many common machine learning tasks.

Figure 3 shows the most important parts of the schema syntax (eliding some details, such as defaults, integers, and array schemas, that are not needed for examples in this paper). The start symbol, *schema*, can be an enumeration, a range, an object, or it can nest other schemas by using one of three logic connectives. An *enum* schema specifies a categorical hyperparameter with a choice between one or more values. Figure 4 includes multiple examples of enums: [*mle*], [*true*, *false*], [0.25], [*linear*, *sag*, *lbfgs*], etc. A *range* schema specifies a continuous hyperparameter with a minimum and

```
schema ::= enum | range | object | disj | conj | neg
enum   ::= [ VALUE⁺ ]
range  ::= ( NUM .. NUM ) distr?
distr  ::= "uniform" | "loguniform"

object ::= dict { (NAME : schema)* }
disj   ::= schema ∨ schema
conj   ::= schema ∧ schema
neg    ::= ¬ schema
```

Figure 3: Schema syntax.

$PCA$ : $\text{dict}\{N\!:(0..1) \vee [mle]\}$
$J48$  : $\text{dict}\{R\!:[true, false], C\!:(0..0.5)\}\wedge$
$\quad\quad (\text{dict}\{R\!:[true]\} \Rightarrow \text{dict}\{C\!:[0.25]\})$
$LR$   : $\text{dict}\{S\!:[linear, sag, lbfgs], P\!:[l1, l2]\}\wedge$
$\quad\quad (\text{dict}\{S\!:[sag, lbfgs]\} \Rightarrow \text{dict}\{P\!:[l2]\})$

Figure 4: Schema examples for 3 operators (simplified): sklearn's `PCA` and `LogisticRegression` and Weka's `J48`. We write $s \Rightarrow t$ for $(\neg s) \vee t$.

maximum (most AutoML optimizers do best with bounded ranges) and an optional distribution *distr*. Figure 4 includes two examples of ranges: $(0..1)$ and $(0..0.5)$. An *object* schema specifies a dictionary of zero or more hyperparameter schemas. For example, Figure 4 specifies one hyperparameter for the PCA operator from sklearn [7] and two each for J48 from Weka [18] and for LR from sklearn.

The schema syntax supports three logic connectives disjunction ($\vee$), conjunction ($\wedge$), and negation ($\neg$). Disjunction describes a union type; for example, in Figure 4, the $N$ hyperparameter of PCA can specify the number of components with either a range $(0..1)$ or an enum $[mle]$. Conjunction describes an intersection type and negation can be used to describe an implication. These connectives help express *constraints* for ruling out combinations of hyperparameters that are individually valid but collectively invalid. For example, with Weka's J48 operator, if $R$ is *true* (use reduced error pruning), then $C$ must be 0.25 (cannot set the pruning confidence to other values). Figure 4 specifies this via $\Rightarrow$, which is syntactic sugar for the basic logic connectives. Hyperparameter constraints are useful for pruning invalid areas from the search space, so AutoML search can focus on valid areas instead.

To make schemas easier to use, we decided not to invent our own new schema language, but instead to adopt a mature existing language with a broad user base and tooling ecosystem. One choice we considered was Python 3 types [45, 37]. Unfortunately, Python 3 types are not expressive enough for specifying constraints. But fortunately, there is another language that also works well with Python and is used extensively by a large community well beyond that of AutoML: JSON Schema [35]. Most constructs from Figure 3 have exact equivalents in JSON Schema. For instance, a range $(a..b)$ can be written as `{"type": "number", "minimum":`$a$`, "maximum":`$b$`}`. Lale adds a few extensions to JSON Schema. For instance, it supports an optional `"distribution"` keyword in a `"number"` schema; and it supports an `"operator"` type for hyperparameters that accept a nested machine-learning operator.

## 3 Gradual Automation

The previous section described pipeline combinators and hyperparameter schemas. This section builds on that foundation to describe a programming model for gradual AutoML. Here, *gradual* means that users get to use total automation if they so desire, but can also exercise fine-grained manual control over any aspects of the pipeline. The programming model design was guided by three principles. First, *progressive disclosure* lets users start by learning just a subset of the constructs, only requiring them to learn more constructs as needed to accomplish more tasks. Second, *orthogonality* minimizes the number of independent constructs to make the programming model easy to learn, while maximizing the ways these constructs can be plugged together to make the programming model expressive [27]. Third, the *principle of least surprise* makes each construct look and behave the way most users would expect. That is why we started from well-established programming models sklearn [7] and JSON Schema [35] and try to be faithful to their expectations.

Figure 5 show the learning curve of gradual AutoML from zero to mastery. The rest of this section describes each set of constructs, starting at the bottom and working upwards one layer at a time.

**Use total automation.** Lale provides an `AutoPipeline` operator. From the outside, it looks and feels just like any sklearn operator with `fit` and `predict` methods. Internally, `fit` runs AutoML on a predefined planned pipeline, which users need not dwell on. Hence, total automation with Lale is trivial to learn for users who already know sklearn (least surprise).

**Inspect returned pipeline.** One level up from total automation is to look at its result: even users who have little interest in controlling the pipeline may want to see it for transparency. Lale provides a `visualize` method and a round-trippable `pretty_print` method that returns a pipeline's Python code.

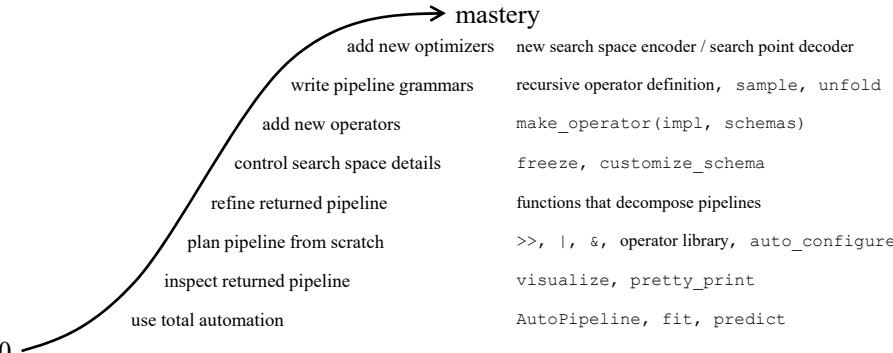

Figure 5: Gradual automation with progressive disclosure.

**Plan pipeline from scratch.** Learning about the combinators and the `auto_configure` method from Section 2 brings users to the next level. To further assist in using Lale's collection of 216 operators, Lale generates documentation from schemas and uses schemas for error checking [17, 38].

**Refine returned pipeline.** Data science, even with automation, is often iterative. After an AutoML search (whether with full automation or with a hand-planned pipeline), users may want to tweak the pipeline for another search. Lale supports this with functions for breaking down a pipeline into smaller pieces, which can then be composed back into a new pipeline using combinators (orthogonality).

**Control search space details.** To exercise even more control in a localized and composable way, users can disable AutoML search for a sub-pipeline with `freeze_trainable`, or they can disable refitting with `freeze_trained`. Furthermore, users may want to locally tweak the schema for some hyperparameters of an operator, such as adjusting a range. Given an operator `A`, calling `A.customize_schema(..)` returns a new operator that is the same as `A` except for the given schema tweaks. Again, these constructs are orthogonal to combinators: frozen or customized operators can be used like any others.

**Add new operators.** Given an operator without a schema, Lale infers a simple schema that just sets all hyperparameters to their defaults. Users can then call `customize_schema` to make them selectively tunable. Or they can write a comprehensive schema and attach it to the operator using `make_operator`.

**Write pipeline grammars.** Some AutoML tools [31, 10, 12] explore not a rigid template for pipeline topologies but a possibly unbounded space of topologies, including some that define a context-free grammar using Lale's combinators [20, 28]. Lale lets users define such search spaces by declaring *recursive operators*, which are self-referential definitions (e.g., `g.p = g.p >> q`). Recursive operators are orthogonal to the other constructs such as combinators, with which they can nest. The methods `sample` and `unfold` turn these into normal Lale pipelines suitable for `auto_configure`.

**Add new optimizers.** Only few users get so close to mastery that they desire to add their own optimizers. While the details will vary on a case-by-case basis, the next section describes a principled translation scheme that helped Lale target several existing optimizers.

## 4 Translation Scheme

This paper adds a third execution mode to sklearn-style pipelines: the first two modes are prediction and training, and the third mode is AutoML search. Lale's API for this mode is `auto_configure`, which first translates the pipeline to a search space for a given optimizer and then runs that optimizer.

Figure 6 details that interaction to illustrate how the translation scheme (encode and decode) interacts with an optimizer and how it uses the first two execution modes (predict and fit). The starting point is a *planned* pipeline, which has operators whose hyperparameters need to be tuned or operator choices (`|` combinator) whose operators need to be selected. Step "encode search space" translates the planned pipeline to an optimizer-specific search space. Once called, the optimizer loops over trials, which are search points it selects based on its solver, e.g., TPE [5]. Step "decode search point" turns an optimizer-specific *search point* (i.e., a specific value combination in the search space) into the configuration needed for instantiating a trainable pipeline. In a *trainable* pipeline, all hyperparameters

and operator choices are bound to specific values. Fitting the trainable pipeline to training data (folds) yields a *trained* pipeline, where the learnable coefficients of all operators are also bound. The next step is to call predict on the trained pipeline and to score the result to compute a loss for the optimizer. After wrapping up its loop, the optimizer returns the best trial, and `auto_configure` returns the corresponding trained pipeline.

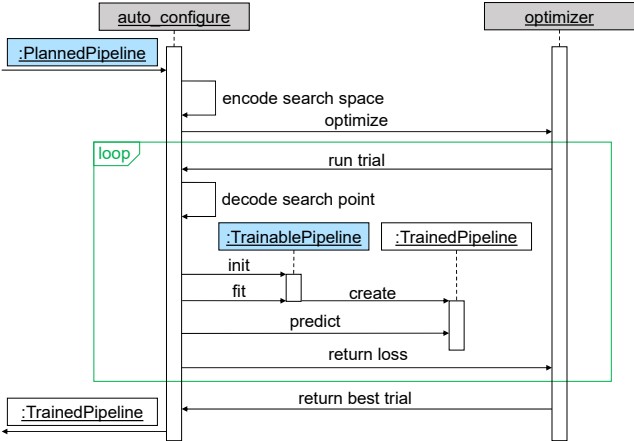

Figure 6: Interaction with an optimizer.

The translation scheme is one of the contributions of this paper. Its job is to enable the modular combinator-based syntax for users, by weaving together the operators and their hyperparameter schemas and rewriting them to a form suitable for a given optimizer. Users can thus make local changes using orthogonal constructs without worrying about their impact on the rest of the pipeline. To target a variety of optimizers, we structured the translation scheme as a series of rewriting *passes* between different *forms*. Each form has its own syntax restricting how a search space can be expressed. Figure 3 shows the starting form. Each pass is designed to be semantics-preserving, in the sense that the search space encodes the same set of pipeline instances before and after the pass. The running example for this section is the pipeline `PCA >> (J48 | LR)`; Figure 4 shows its schemas in starting form.

**Normalize.** Figure 7 shows the normal form, which is a disjunction of objects. This form is crafted to simplify later rewriting passes by letting them start from a less feature-laden form. While nesting is still possible in normal form, it is restricted to objects. Also, once in normal form, schemas contain no conjunctions or negations, which would inhibit local reasoning. Normalization works in a bottom-up pass. At each level, it performs disjunction hoisting using rewrites such as De-Morgan's law. Negation and conjunction are pushed into primitive schemas. This can temporarily give rise to uninhabited or trivial nested schemas, which the pass simplifies away with another set of rewrites. Our translation scheme normalizes the schema of each operator in a pipeline separately.

$$
\begin{array}{ll}
schema ::= object\ (\vee\ object)^{\star} \\
object\ ::=\ \texttt{dict}\ \{\ (NAME : prop)^{\star}\ \} \\
prop\ \ ::=\ enum\,|\,range\,|\,object \\
enum\ \ ::=\ [\ VALUE^{+}\ ] \\
range\ \ ::=\ (\ NUM\,..\,NUM\ )\ distr^{?} \\
distr\ \ ::=\ \texttt{"uniform"}\ |\ \texttt{"loguniform"}
\end{array}
\qquad
\begin{array}{ll}
PCA : \text{dict}\{N\!:\!(0..1)\} \vee \text{dict}\{N\!:\![mle]\} \\
J48\ \ : \text{dict}\{R\!:\![false], C\!:\!(0..0.5)\}\vee \\
\qquad \text{dict}\{R\!:\![true, false], C\!:\![0.25]\} \\
LR\ \ \ : \text{dict}\{S\!:\![linear], P\!:\![l1, l2]\}\vee \\
\qquad \text{dict}\{S\!:\![linear, sag, lbfgs], P\!:\![l2]\}
\end{array}
$$

Figure 7: Normalized schemas: syntax (left) and example (right).

**Combine.** Figure 8 shows the combined form, which represents the entire pipeline together instead of each operator separately. Combining encodes combinators as follows. The `>>` and `&` combinators turn into an object schema (dict), where each operator becomes a step $s_i$. The `|` combinator turns into a disjunction schema ($\vee$), with an additional discriminant $D$ to track what was chosen. Higher-order operators yield nested object schemas (dict). The encoding preserves sufficient information to enable later decoding. After this pass, the encoding is conceptually similar to that of Hyperopt [6], so our implementation provides a simple backend translating it into a Hyperopt search space.

**Flatten.** Most other optimizers require a less hierarchical search space than the combined form. Figure 9 shows the flattened form, which is a disjunction of objects without further nesting. The flatten pass hoists properties from nested objects, renaming them as needed for uniqueness (in a way that enables later decoding). The flattened form is close to what SMAC [15] needs, except that objects of different disjuncts have different properties. In SMAC, all disjuncts have the same properties, with

$$\begin{aligned}
&space &::= &\ object\\
&object &::= &\ \texttt{dict}\ \{\ (NAME : prop)^{\boldsymbol{*}}\ \}\\
&prop &::= &\ enum \mid range \mid disj\\
&disj &::= &\ object\ (\vee\ object)^{\boldsymbol{*}}
\end{aligned}
\qquad
\text{dict}\left\{
\begin{array}{l}
s_0\!: \text{dict}\{N\!:\!(0..1)\} \vee \text{dict}\{N\!:\![mle]\}\\[4pt]
s_1\!: \left(
\begin{array}{l}
\left(\begin{array}{l}
\text{dict}\{D\!:\![J48], R\!:\![false], C\!:\!(0..0.5)\} \vee\\
\text{dict}\{D\!:\![J48], R\!:\![true,false], C\!:\![0.25]\}
\end{array}\right) \vee\\
\left(\begin{array}{l}
\text{dict}\{D\!:\![LR], S\!:\![linear], P\!:\![l1,l2]\} \vee\\
\text{dict}\{D\!:\![LR], S\!:\![linear,sag,lbfgs], P\!:\![l2]\}
\end{array}\right)
\end{array}\right)
\end{array}\right\}$$

Figure 8: Combined search space: syntax (left) and example (right).

$$\begin{aligned}
&space &::= &\ object\ (\vee\ object)^{\boldsymbol{*}}\\
&object &::= &\ \texttt{dict}\ \{\ (NAME : prop)^{\boldsymbol{*}}\ \}\\
&prop &::= &\ enum \mid range
\end{aligned}$$

$$\begin{array}{llll}
&\text{dict}\{N\!:\!(0..1),&D\!:\![J48],&R\!:\![false],&C\!:\!(0..0.5)\}\\
\vee&\text{dict}\{N\!:\!(0..1),&D\!:\![J48],&R\!:\![true,false],&C\!:\![0.25]\ \}\\
\vee&\text{dict}\{N\!:\![mle],&D\!:\![J48],&R\!:\![false],&C\!:\!(0..0.5)\}\\
\vee&\text{dict}\{N\!:\![mle],&D\!:\![J48],&R\!:\![true,false],&C\!:\![0.25]\ \}\\
\vee&\text{dict}\{N\!:\!(0..1),&D\!:\![LR],&S\!:\![linear],&P\!:\![l1,l2]\ \}\\
\vee&\text{dict}\{N\!:\!(0..1),&D\!:\![LR],&S\!:\![linear,sag,lbfgs],&P\!:\![l2]\ \ \ \ \}\\
\vee&\text{dict}\{N\!:\![mle],&D\!:\![LR],&S\!:\![linear],&P\!:\![l1,l2]\ \}\\
\vee&\text{dict}\{N\!:\![mle],&D\!:\![LR],&S\!:\![linear,sag,lbfgs],&P\!:\![l2]\ \ \ \ \}
\end{array}$$

Figure 9: Flattened search space: syntax (left) and example (right).

conditionals on the side indicating which variables are relevant. The SMAC backend thus generates these conditionals, and renders the space in SMAC's parameter configuration space format [42].

The ADMM backend uses a partially flattened form. ADMM [26] supports a pipeline of operator choices. The backend thus leaves the top-level spine of a pipeline intact as a pipeline of choices. If one of the choices contains a complex operator (e.g. a nested pipeline), it is flattened as for SMAC.

**Discretize.** The optimizers GridSearchCV and HalvingGridSearchCV from sklearn [7] require continuous ranges to be made categorical. Discretization for a hyperparameter first includes the default and then samples a user-configurable number of additional values from the range based on the prior distribution. Figure 10 shows the discretized form when generating two values per range.

$$\begin{aligned}
&space &::= &\ object\ (\vee\ object)^{\boldsymbol{*}}\\
&object &::= &\ \texttt{dict}\{\ (NAME : enum)^{\boldsymbol{*}}\ \}
\end{aligned}$$

$$\begin{array}{l}
\text{dict}\{N\!:\![0.50, 0.01], D\!:\![J48], R\!:\![false], C\!:\![0.25, 0.01]\}\\
\vee\ \text{dict}\{N\!:\![0.50, 0.01], D\!:\![J48], R\!:\![true,false], C\!:\![0.25]\}\\
\vee\ \text{dict}\{N\!:\![mle], D\!:\![J48], R\!:\![false], C\!:\![0.25, 0.01]\}\\
\vee\ ...\\
\vee\ \text{dict}\{N\!:\![mle], D\!:\![LR], S\!:\![linear,sag,lbfgs], P\!:\![l2]\}
\end{array}$$

Figure 10: Discretized search space: syntax (left) and example (right).

## 5   Results

This section presents experiments evaluating Lale's usability and expressiveness. While the experiments focus on tabular data and do not use deep learning, Lale also works with other data modalities such as text [8], code [29], and time-series [39], including with pipelines that contain deep-learning operators.

**RQ1: How usable is gradual AutoML with combinators?**

To explore this research question, we compare Lale to the popular sklearn machine learning framework in a user study with 18 participants. The study design is between-subjects [9], where participants are randomly assigned to a Lale or sklearn version of the study. We asked participants to perform four tasks involving sample machine learn-

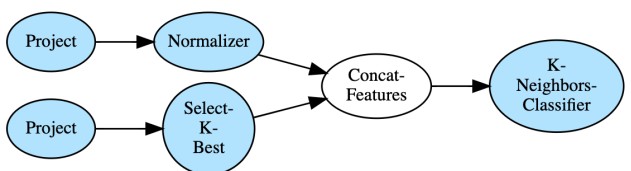

Figure 11: Sample pipeline used in user studies.

ing pipelines: (T1) inspecting and understanding an existing pipeline, (T2) refining the given pipeline by replacing operators and hyperparameters, (T3) debugging a failing pipeline, and (T4) searching across a space of potential pipelines. The tasks are based on the principle of *progressive disclosure* in that we provide a sample pipeline (Figure 11) using common manual constructs in either sklearn or

Lale format and ask participants to perform tasks that use a progressively expanding set of constructs. All participants self-reported having at least some sklearn experience, worked at IBM, and were not further compensated for their voluntary participation. The supplemental material describes the detailed study design and methodology, including the sklearn and Lale notebooks used.

One surprising result of the user study is that four (44%) of the sklearn participants were not able to correctly identify the branching structure of the preprocessing subpipelines of the sample pipeline during T1 (seen in Figure 11), compared to a single Lale participant (11%). Two of the Lale participants also used inspection features in Lale such as the `visualize` and `pretty_print` methods. In T4, two (22%) of the sklearn participants were either unable to correctly (or at all) implement some method of searching across a space of potential pipelines whereas all of the Lale participants were able to complete the task. When implementing T4, most of the Lale participants (6 or 67%) used the *or* combinator (`|`) and the AutoML execution mode to perform automated search as opposed to more manual methods. This is compared to two (22%) sklearn participants using automated search via GridSearchCV. The relatively high success rate of Lale participants both in understanding existing pipelines that use the *and* combinator (`&`) and in composing new planned pipelines that use the *or* combinator (`|`) suggests that new Lale users are able to pick up and use these independent constructs. This supports the principle of *orthogonality* in the Lale programming model. Automated search also tends to be more concise than manual implementations for the search task, perhaps contributing to Lale participants on average implementing the task in 25 fewer lines of code (LoC) (59% less) compared to sklearn participants. Lale participants also on average performed all of the tasks faster than sklearn participants, over 2.5 minutes faster total (13% faster). (In reality, the time difference would likely be even higher, since we stopped participants who take a manual approach early, see supplement for details.) For T2 and T3, participants performed equally well in both versions of the study. We note that some Lale participants were able to refine pipelines without referring to documentation, relying on their sklearn knowledge to replace operators and define their hyperparameters. This supports the *principle of least surprise* in that Lale operators work in ways that experienced sklearn users expect. Table 1 summarizes the results from the user study.

Table 1: Summary of user study results.

| VERSION | COUNT | T1 CORRECT | T4 CORRECT | T4 LoC MEDIAN (ALL) | TOTAL TIME MEAN (STDDEV) |
|---------|-------|-----------|-----------|---------------------|--------------------------|
| Lale | 9 | 89% | 100% | 10 (7, 8, 9, 9, 10, 10, 10, 14, 75) | 17:36 (5:03) |
| sklearn | 9 | 56% | 78% | 23 (12, 15, 16, 22, 24, 43, 100, 100) | 19:26 (6:34) |
| Total | 18 | 72% | 89% | 14 | 18:54 (5:45) |

*Answer to RQ1:* In our user study, gradual AutoML with combinators was usable enough to increase task correctness by more than 20% while reducing lines of code by more than half.

**RQ2: Does the translation scheme work for diverse planned pipelines?**    To explore this research question, we used Lale's `auto_configure` on several planned pipelines with Hyperopt and its TPE solver. We used auto-sklearn [15] as a state-of-the-art baseline; it uses SMAC for joint algorithm selection and hyperparameter tuning similar to our work, but also uses meta-learning, which we do not yet do. We used a 2.0GHz virtual machine with 32 cores and 128GB memory and gave each search a 1 hour time budget with a timeout of 6 minutes per trial, which corresponds to the default setting of auto-sklearn. We chose 14 datasets from OpenML [46] (CC-BY license) that allow for meaningful optimization (as opposed to just the initial few trials) within that 1-hour budget. Our datasets are drawn from the AutoML Benchmark [16] and four of them were also used in the auto-sklearn paper [15]. (The goal of our evaluation is orthogonal to the sizes of the datasets.) We used a 66:33% train:test split with 5-fold cross validation on the train set during optimization.

Table 2 summarizes the results. Column AUTOSKL shows the auto-sklearn baseline; LALE-AUTO uses Lale's `AutoPipeline` operator from Section 3; LALE-AD3M and LALE-TPOT use Lale grammars inspired by AlphaD3M [12] and TPOT [31] unfolded with a maximal depth of 3; LALE-ADB uses a pipeline with a choice of preprocessors and the `AdaBoostClassifier` higher-order operator; and ASKL-ADB is auto-sklearn [15] with the set of classifiers limited to AdaBoost with only preprocessing. (See supplement for details.) All searches succeed and most yield performance in the same ballpark as auto-sklearn (aside from shuttle, which triggers an auto-sklearn issue that other users have also encountered). Considering all 14 datasets, LALE-AUTO performs on average 0.7% better than AUTOSKL. Excluding shuttle, Lale performs on average 0.3% worse than AUTOSKL. These 0.3% are smaller than the standard deviation on most datasets, and based on a t-test, they are not statistically significant.

Table 2: Test accuracy for 14 OpenML classification tasks, using 4 Lale search spaces with Hyperopt.

| DATASET | Absolute accuracy mean (and stddev) over 5 runs | | | | | | $100 * (\text{LALE}/\text{AUTOSKL} - 1)$ | | | |
| | AUTOSKL | LALE-AUTO | LALE-TPOT | LALE-AD3M | LALE-ADB | ASKL-ADB | AUTO | TPOT | AD3M | ADB |
|---|---|---|---|---|---|---|---|---|---|---|
| australian | 85.1(0.4) | 86.2 (0.0) | 85.9 (0.6) | 86.8 (0.0) | 86.0 (1.6) | 84.7 (3.1) | 1.3 | 0.9 | 2.0 | 1.1 |
| blood | 77.9(1.4) | 75.3 (0.0) | 77.5 (2.5) | 74.7 (0.7) | 77.1 (0.7) | 74.7 (0.8) | -3.3 | -0.5 | -4.0 | -1.0 |
| breast-cancer | 73.0(0.6) | 73.0 (0.0) | 71.4 (1.1) | 69.5 (3.3) | 70.9 (2.0) | 72.4 (0.5) | 0.0 | -2.3 | -4.9 | -2.9 |
| car | 99.4(0.1) | 97.7 (0.0) | 99.1 (0.1) | 92.7 (0.6) | 98.3 (0.3) | 98.2 (0.2) | -1.6 | -0.2 | -6.7 | -1.1 |
| credit-g | 76.6(1.2) | 75.7 (0.0) | 74.1 (0.5) | 74.8 (0.4) | 76.1 (1.3) | 76.2 (1.0) | -1.1 | -3.2 | -2.4 | -0.7 |
| diabetes | 77.0(1.3) | 76.3 (0.0) | 76.4 (1.1) | 77.9 (0.2) | 76.0 (0.5) | 75.0 (1.0) | -0.9 | -0.8 | 1.1 | -1.3 |
| jungle-chess | 88.1(0.2) | 92.4 (0.0) | 88.9 (2.0) | 74.1 (2.0) | 89.4 (2.3) | 86.9 (0.2) | 4.9 | 0.9 | -15.8 | 1.5 |
| kc1 | 83.8(0.3) | 83.4 (0.0) | 83.5 (0.5) | 83.6 (0.2) | 83.3 (0.4) | 84.0 (0.3) | -0.5 | -0.4 | -0.2 | -0.6 |
| kr-vs-kp | 99.7(0.0) | 99.5 (0.0) | 99.4 (0.0) | 96.8 (0.1) | 99.5 (0.1) | 99.5 (0.2) | -0.2 | -0.3 | -2.9 | -0.2 |
| mfeat-factors | 98.7(0.1) | 97.1 (0.0) | 97.9 (0.5) | 97.5 (0.1) | 97.5 (0.4) | 97.9 (0.1) | -1.6 | -1.5 | -1.2 | -1.2 |
| phoneme | 90.3(0.4) | 89.5 (0.0) | 89.6 (0.4) | 76.6 (0.0) | 90.1 (0.4) | 91.4 (0.2) | -0.8 | -0.8 | -15.2 | -0.2 |
| shuttle | 87.3(11.6) | 100.0 (0.0) | 99.9 (0.0) | 99.9 (0.0) | 100.0 (0.0) | 100.0 (0.0) | 14.5 | 14.5 | 14.4 | 14.6 |
| spectf | 87.9(0.9) | 87.7 (0.0) | 88.4 (2.2) | 83.6 (6.9) | 88.4 (2.6) | 89.7 (2.9) | -0.2 | 0.6 | -4.9 | 0.6 |
| sylvine | 95.4(0.2) | 95.0 (0.0) | 94.4 (0.7) | 91.3 (0.1) | 95.1 (0.2) | 95.1 (0.1) | -0.4 | -1.1 | -4.3 | -0.3 |

*Answer to RQ2:* The translation scheme works for pipelines with complex topologies, including those generated by grammars inspired by AlphaD3M and TPOT, as well as pipelines including ensembles, producing competitive results.

**RQ3: Does the translation scheme work for diverse optimizer backends?** To explore this research question, Table 3 lists test accuracies using 9 different optimizers for the OpenML phoneme dataset. The planned pipeline uses a scaler, a polynomial feature generator, and a choice between XGBoostClassifier and LGBMClassifier. The other experimental settings are the same as above. All translations succeed, and most yield reasonable accuracy, except for the ones that timed out. The translated search spaces for GridSearchCV and HalvingGridSearchCV were correct but too large to explore within the time budget. Most people would only use grid-based search for very small search spaces, but then, suffer from exploring the space less thoroughly than "proper" AutoML tools.

Table 3: Test accuracy (mean and stddev) for the phoneme dataset using multiple optimizer backends.

| | Hyperopt | | Hyperband | | ADMM | | GridSearchCV | Halving- |
| TPE | Anneal | Rand | | RND | BOBa | GPRND | | GridSearchCV |
|---|---|---|---|---|---|---|---|---|
| 88.75 (0.2) | 88.94 (0.3) | 88.75 (0.5) | 87.94 (0.2) | 89.00 (0.2) | 89.14 (0.2) | 88.61 (0.3) | Timed out | Timed out |

*Answer to RQ3:* The translation succeeds for Hyperopt (3 optimizers), Hyperband, ADMM (3 optimizers), GridSearchCV, and HalvingGridSearchCV.

# 6 Related Work

Combinators, which were discovered in the 1920s [41], have long been prominent in functional programming languages such as Lisp [43]. The advantage of their tacit style has been recognized more broadly; for instance, the Unix pipe [30] can be viewed as a combinator. Combinators have also enabled some sophisticated innovations such as parser combinators [23].

**Combinators and data science.** MapReduce [11] is a successful programming model for large-scale data processing named after two combinators [22]. Allison [2] describes Haskell combinators for converting between different machine-learning model types, but does not discuss AutoML. Sklearn [7] provides combinators (`make_pipeline` and `make_union`) as well as some basic AutoML (`GridSearchCV`, `HalvingGridSearchCV`, and `RandomizedSearchCV`) but only integrates combinators with AutoML loosely, instead requiring users to perform manual name-mangling and non-compositional repetition. Lale builds on sklearn, extending and streamlining its combinators and tightly integrating them with AutoML. Pilat et al. [36] describe a combinator language suitable as a target for automatic code generation with genetic programming, but not intended to be written by hand. Unlike our work, they focus on total (not gradual) automation and do not provide an *or* combinator ( | ).

**AutoML programming models.** Auto-WEKA demonstrated how to support conditional hyper-parameters on multiple solvers, but the paper focused on a single planned pipeline, not gradual automation [44]. While both hyperopt-sklearn [21] and auto-sklearn [15] support some gradual automation, instead of tightly integrating gradual automation with combinators, they require users to

drop to a bespoke lower-level API: Hyperopt for hyperopt-sklearn and SMAC's parameter configuration space format [42] for auto-sklearn. We argue that JSON Schema [35], being a language with a track record of wide-spread use by programmers who are not AutoML experts, is less low-level; e.g., it is the backbone of OpenAPI [32]. We also built a tool to extract schemas from sklearn docstrings [4]. In Optuna, users write imperative code that first samples values, then uses either if-statements for operator choice or passes values to constructors for hyperparameter tuning [1]. In contrast, combinators are more concise, more localized, and less surprising to sklearn users. OBOE [48] proposes an alternative meta-learning approach from that of auto-sklearn; it requires an initial discrete set of models with hyperparameters to choose from, and Lale could potentially be used to specify this set of models. In PyGlove, users start from a manually written neural network which they then "hyperify" by replacing a component or hyperparameter with a "hyper value" [34]. Their notion of "hyper" is similar to our notion of "planned", but they do not use combinators, and they focus only on neural architecture search. AutoGluon-Tabular focuses on total automation and the paper does not discuss gradual automation [13]. In AutoGOAL, users specify a hierarchy of pre-defined "indicator" types, then the system tries out any pipeline of operators that gets from the input to the output [14]. AutoGOAL's API deviates from sklearn conventions, and while it uses the syntax of Python 3 types, it does not capture constraints, nor is it compatible with Python 3 type checkers.

## 7    Limitations and Societal Impacts

A limitation of the user study is that it used only 18 participants (9 for the sklearn treatment plus 9 different participants for Lale), and that all participants worked at IBM. Another potential limitation is that researcher bias may be present in the design of the user study tasks. We designed the tasks to showcase and validate aspects of the Lale programming model which may potentially bias tasks in favor of Lale. We attempt to mitigate this by only selecting participants who are experienced in sklearn and by only allowing Lale newcomers to participate in the Lale version of the study.

One limitation of the translation scheme is that it is not guaranteed to handle all JSON Schema features; in fact, we encountered and fixed some limitations early during development. Fortunately, there were no new limitations recently, and there is theoretical work that could be used to tackle them if they arise [3]. A limitation of schema-based constraints is that they currently do not capture interdependence between operators. We mitigate that by providing not just hyperparameter schemas but also dataset schemas, which enable us to check whether data flowing from one operator to another in a pipeline is compatible [17]. A limitation of the programming model is that, while it achieves a high level of sklearn compatibility, Python code that explicitly reflects over dynamic types can still expose differences. We mitigate this by testing that Lale interoperates well with a broad set of sklearn features including metrics, cross-validation, operators, etc.

One potential societal impact is that enabling automation of searching machine learning pipelines may encourage computationally-heavy approaches to data science tasks and raise $CO_2$ emissions. On the other hand, we hope gradual automation will encourage data scientists to use their domain knowledge to more selectively and efficiently automate their machine learning work. In future work, we are planning to adopt caching techniques [24, 47] to further reduce $CO_2$ emissions.

## 8    Conclusion

This paper presents combinators for gradual AutoML, along with a novel translation scheme from combinators to existing optimizers. We have implemented our combinators in an open-source library called Lale, while ensuring they interoperate smoothly with the sklearn ecosystem. Besides being open-source, Lale is also actively used in a commercial AutoML product, highlighting the real-world value of the underlying concepts. Our ultimate goal is to make AutoML as accessible as manual ML by offering a gradual combination of both.

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
