# OpenReview forum: "Pipeline Combinators for Gradual AutoML"
_NeurIPS.cc/2021/Conference — NeurIPS 2021 Poster_

### Official Review · Reviewer_cjD3 · 2021-07-11

**Rating:** 4
**Confidence:** 5

**Summary:**

The authors introduced a wrapper for hyperparameter optimization, named Lale, to create and verify search spaces more easily. To this end, instructions with combinators are used, which further are translated to fit the underlying optimizer (like SMAC, Hyperopt, etc.). Lale is meant to be an interface between machine and human s.t. parts of the pipeline can be easily changed, if desired.

**Limitations And Societal Impact:**

Yes.

**Main Review:**

# Novelty
The novelty part can be narrowed down to the unified translation schema, which is the first work in the domain of AutoML to my knowledge. However, it is not clear to me whether this is more novelty in the sense of engineering some useful software or whether this sufficient scientific novelty in the method directly.

# Significance / Impact
The advantages for Lale over existing AutoML frameworks are especially advantageous for inexperienced users due to two reasons. (i) The unified interface makes it easy to quickly set up an AutoML pipeline for any implemented optimizer. (ii) Graph overviews are a helpful and easy way to check whether the search space is defined correctly.

However, experts in ML and AutoML tend to work as closely to the optimizer interfaces as possible to have full control over the optimizer. Since defining search spaces is transparent and easily understandable in several frameworks too, e.g. SMAC and Hyperopt, Lale would not give many advantages over these. Full AutoML systems such as AutoSklearn, on the other hand, would definitely benefit from Lale. Overall, I’m not fully convinced that there will be many users of Lale at the end of the day.

Furthermore, I don’t see a clear path for scientific future work building on top of Lale.

Overall, it is uncertain whether Lale will have a great impact in the community.

# Soundness (method and experimental setup)

The authors described their method and experiments detailedly and the concept is sound.

I appreciate the overall idea of having several API levels which allow to gradually build up expertise in using AutoML tools. Depending also on the level of the user’s expertise, this should allow to find an appropriate API without switching between different packages. However, I have the impression that conceptually this is a very high-level idea of software engineering and unclear how this scientifically contributes to the NeurIPS community.

# Scholarship
Related work covers both combinators in general and in the AutoML domain. The domain is sufficiently covered.


# Clarity
The authors introduced their AutoML library in a sorted, logical way.

# Minor Comments
* Using Lale still requires a decent background on machine learning. Calling methods and Interpreting graphs (Figure 11) may not be straight forward for inexperienced users.
* As already mentioned in the author’s limitations, the user studies only include 9 participants, making statements not very meaningful. Additionally, the survey might even be  biased towards Lale.
* Figure 6 shows a default routine with the only difference of “encode search space”. There’s no novelty aspect at all.
* Table 1 would not convince me to use Lale over Auto-Sklearn.

# Questions for Rebuttal
* Table 2: It seems like Auto-Sklearn outperforms Lale-Auto. What are the reasons for that?


**Time Spent Reviewing:**

2

---

> ### Author Response · Authors · 2021-08-10
> **Author Response to Reviewer cjD3**
>
> > The novelty part can be narrowed down to the unified translation
> > schema, which is the first work in the domain of AutoML to my
> > knowledge. However, it is not clear to me whether this is more
> > novelty in the sense of engineering some useful software or whether
> > this sufficient scientific novelty in the method directly.
>
> We argue that the novelty of our paper is the programming model
> together with the translation scheme that makes the programming model
> possible.  In your words, one way to view this is as "engineering some
> useful software", which is well-aligned with the call-for-papers
> category "Infrastructure (e.g., datasets, competitions,
> implementations, libraries)". A more scientific way to phrase this is
> the hypothesis "combinators and JSON schemas can effectively specify
> search spaces for AutoML". This hypothesis felt radical at the outset
> of our project, but in retrospect, our answer is a resounding "yes".
>
> > experts in ML and AutoML tend to work as closely to the optimizer
> > interfaces as possible to have full control over the optimizer.
>
> This may be true for people who are experts in both ML and AutoML.
> However, that is a small population, and there are far more people who
> may have hands-on experience in ML and data science but at most a
> working knowledge in AutoML. Lale is designed for this broader
> population.
>
> > Table 2: It seems like Auto-Sklearn outperforms Lale-Auto. What are
> > the reasons for that?
>
> If you look at all datasets, then Lale outperforms auto-sklearn.
> If you exclude the dataset on which auto-sklearn performs worst
> (shuttle), then on the remaining datasets, the average accuracy with
> Lale is 0.3% worse than with auto-sklearn. Note that unlike Lale,
> auto-sklearn benefits from warm-start via meta-learning.
> That said, given that 0.3% is less than the standard deviation on most
> datasets, and given the shuttle result, we did not deem the difference
> important enough to investigate further.

---

### Official Review · Reviewer_H7JW · 2021-07-12

**Rating:** 5
**Confidence:** 3

**Summary:**

This paper propose a new autoML system with the formalization of pipeline combinators. At the high-level, machine learning can be automatically used by novices without any expertise; at the low-level, a formalization is proposed to facilitate the search of optimal hyper parameters. User studies and experiments are included to justify the designs.

**Ethical Concerns:**

Not applicable.

**Limitations And Societal Impact:**

Yes.

**Main Review:**

AutoML is a useful tool to help novices apply machine learning in their domains, thus, to design easy-to-use and efficient autoML system is an interesting research topic. This paper proposes a system with some new formalizations under its software layout. The motivation is clear: to make the utilization of the system as easy as possible while enabling efficient optimization of the hyper-parameters demanded by different machine learning pipelines.

Technically, I have a few comments:
+ I am confused by the design philosophy  of the formalization of combinator in Section 2.  Essentially, why do we need these formalizations? The author keeps mentioning that there are some roughly equivalent operators within the sklearn toolkit. Does this section just offer some more formal definition of a language, or is the syntax defined in this section more expressive than the existing systems? It would be better to have a more explicit description about the motivation of the design.
+ W.r.t the grammar of "pipeline", I am wondering if this could make the formalization be able to be used for neural network architecture search? It seems there is no such discussion.

The presentation of the empirical study should be organized better:
+ For RQ1, even though there is limited number of participants (which is fine due to the difficulty of organizing the user study), there still should be some formal description about the null hypothesis in this user study, then there should be some statistical analysis about the statistical significance of the result, e.g., one can consider bootstrap based methods to report the p-value w.r.t the hypothesis. Further, this section should be self-contained, there should be some brief introduction about the tasks.

+ For RQ2 and RQ3, it seems that only the end-to-end performance boost is reported, on the other hand, it is important to understand where the performance gain comes from (perhaps based on some micro-benchmarks). In other words, the section should answer the question why the formalization of the combinators is effective for the black-box optimizer.

Post rebuttal updates:

I really appreciate the great effort the author has made to address my concerns. On the other hand, I hope during this procedure, the author would also feel that both my comments and comments from other reviewers could help to make it a better paper. Here are some follow-up suggestions:

+ To be more clear about the scientific contribution from the very beginning of the paper, e.g., a more expressive formalization of the AutoML system.

+ To organize the presentation of the empirical study following the general scientific principles; and to recruit more participants if possible so that one could draw some statistically significant conclusions.

+ To provide some analysis about the performance gain and to discuss how the gain relates to the proposed design---at the end of the day, as scientists, we not only want to know if something works, we want to know why it works as well.




**Time Spent Reviewing:**

4

---

> ### Author Response · Authors · 2021-08-10
> **Author Response to Reviewer H7JW**
>
> > a formalization is proposed to facilitate the search of optimal
> > hyper parameters
>
> The goal of our library goes beyond hyperparameter search, but also
> covers the whole spectrum of "gradual AutoML" where users can access
> any point in the spectrum (full automation to partial automation to
> defining arbitrary complex search spaces for directed acyclic
> machine-learning pipelines) with a consistent programming interface.
> And even for expert users who would define complex search spaces
> (which can be done succintly using the syntax facilitated by the
> proposed combinators), the library provides various backends
> (Hyperopt, SMAC) to explore these search spaces regardless of their
> complexity.
>
> > Does this section [Section 2] just offer some more formal definition
> > of a language, or is the syntax defined in this section more
> > expressive than the existing systems?
>
> The syntax is more expressive than existing systems. Specifically, the
> syntax is more expressive than sklearn, since sklearn lacks a choice
> combinator (|), and since Lale allows omitting hyperparameters from
> individual operators. Also, the syntax is more expressive than that of
> most existing AutoML tools, since it supports higher-order operators,
> where hyperparameters passed to an outer operator are themselves inner
> pipelines with optimizable operator choices and hyperparameters of
> their own.
>
> > W.r.t the grammar of "pipeline", I am wondering if this could make
> > the formalization be able to used for neural network architecture
> > search?
>
> Yes, the formalization could also be used for neural network
> architecture search. In fact, we have already created prototypes and
> run experiments using Lale for simple topologies of neural network
> componentry including back-propagation. However, these results are
> still preliminary and too premature to include in this paper.
>
> > For RQ1, even though there is limited number of participants (which
> > is fine due to the difficulty of organizing the user study), there
> > still should be some formal description about the null hypothesis in
> > this user study, then there should be some statistical analysis
> > about the statistical significance of the result (...)
>
> As the reviewer points out, the limited number of participants makes meaningful
> quantitative analysis difficult. However, we will provide the following null hypotheses
> to test: 1) T1 correct is the same between Lale and Sklearn, 2) T4 correct is the same,
> 3) T4 lines of code (LoC) is the same, and 4) total time taken is the same. The
> Mann-Whitney test T4 LoC rejects the third null hypothesis with
> a p-value of 0.0039. We are unable to reject the other null hypotheses. This is probably
> due to the small sample size that is common for laboratory-style user studies. When
> bootstrapping the user study data to 100 observations for each task, all four null
> hypotheses are rejected.
>
> > Further, this section should be self-contained, there
> > should be some brief introduction about the tasks.
>
> We agree that there should be brief introductions about the tasks and note our
> descriptions for the tasks starting on line 257. Given the page limit, more detailed
> descriptions would be difficult to include in the main body but note that detailed
> descriptions along with the actual test notebooks are available in the supplemental
> material.
>
> > For RQ2 and RQ3, it seems that only the end-to-end performance boost
> > is reported, on the other hand, it is important to understand where
> > the performance gain comes from (perhaps based on some
> > micro-benchmarks).
>
> Given the page limit, such detailed experiments would be difficult to
> include in the main body of the paper. In fact, earlier versions of
> this paper did include additional performance results, for instance
> with other data modalities besides tabular data.  Based on your
> feedback, we may add more drill-down experiments to the supplemental
> material.

---

> ### Author Response · Authors · 2021-09-02
> **Author response to updates towards the end of the discussion period.**
>
> Thank you for updating your review after our author response!
>
> You wrote:
>
> > I hope during this procedure, the author would also feel that both
> > my comments and comments from other reviewers could help to make it
> > a better paper.
>
> We appreciate the time, effort, and constructive comments from you and
> the other reviewers and will try to use them to make our paper better.

---

### Official Review · Reviewer_G6BM · 2021-07-17

**Rating:** 8
**Confidence:** 3

**Summary:**

This paper aims at constructing a system for gradual AutoML that is concise, modular (or compositional), and easy-to-use. To this end, the paper introduces three orthogonal combinators (i.e., higher-order functions) which enable compositional code for gradual AutoML, and hyperparameter schemas which describe search spaces of hyperparameters. To support various backend AutoML optimizers, the paper proposes a translation scheme which translates pipelines, described by combinators and hyperparameter schemas, into search spaces for those optimizers. The paper implements these ideas into a Python library (called Lale) which includes a new execution mode (called AutoML search) for running AutoML searches. Through user studies and experiments, the paper shows that Lale is easy-to-use, and can express various pipelines and support various optimizers.

**Limitations And Societal Impact:**

Limitations and societal impact are discussed in the paper.

**Main Review:**

I really enjoyed reading this paper! The paper is extremely well-written and well-organized---it was easy to understand both low-level details and high-level ideas clearly. In particular, the paper succeeds at abstracting out unimportant details and explaining core ideas succinctly. Moreover, the technical contents of the paper look novel and sound to me.

The paper considers a very important problem and provides a very satisfactory solution both in technical and practical point of view. Technically, the paper introduces three combinators important for compositional gradual AutoML, among which the or combinator is particularly novel. Also the entire system (including combinators, hyperparameter schema, and translation scheme) seems to be designed and engineered in a principled way, and the paper describes some important design principles. Practically, learning and using Lale look easy to me (which is confirmed by user studies) in that it is implemented in Python and based on popular frameworks (sklearn and JSON Schema). Also Lale supports various pipeline structures and optimizer backends. For these reasons, I think Lale will have a huge positive impact on the NeurIPS community.

Questions & Comments:
- Line 357 describes “dataset schemas” but I think it is not explained elsewhere in the paper. What is it in detail?
- Line 291. “participants where able” --> “participants were able”.

-----
**Updates after the author response.** Thank the authors for answering my questions. I am still fond of the paper and will keep the same score.

----
**Updates after internal discussions.** I like the paper for two reasons (which I wrote in the internal discussion):

* First, I do think the paper has scientific contributions. Most importantly, the paper identifies three combinators (pipe, and, or) for the domain of AutoML, and shows that they are enough to express most tasks in AutoML and further enable "gradual" AutoML by making a programming system for AutoML "modular"---note that the combinators are the key to the modularity. I think this is an important scientific contribution, and would be useful for the NeurIPS community as it enables gradual AutoML (which has not been possible in existing AutoML systems).
* Second, I have made the following assumption: engineering software that is useful to the NeurIPS community is considered as one of the contributions NeurIPS appreciates. In this context, the paper makes an another contribution by providing the Lale library which I think would be quite useful at least for beginners of AutoML. Of course, my above assumption could be wrong and I totally understand if others disagree with it.

However, during internal discussions, other reviewers expressed different thoughts and raised concerns on each of the two points. Given this, I suggest the authors address those concerns in the next version of the paper to make the paper stronger, e.g., by highlighting the first point more and providing concrete/quantitative evidence on how much useful the Lale library is (and will be) for the NeurIPS community.

**Time Spent Reviewing:**

6

---

> ### Author Response · Authors · 2021-08-10
> **Author Response to Reviewer G6BM**
>
> Thank you very much for your review! We hope you can convince the
> other reviewers with your positive opinion of our paper.
>
> > Line 357 describes "dataset schemas" but I think it is not explained
> > elsewhere in the paper. What is it in detail?
>
> Lale uses JSON schema to describe datasets, including the schema of
> data that an operator expects as input to various methods (fit,
> predict, predict_proba, etc.) or produces as output from various methods.
> Typically, this ends up being a (possibly nested) array schema, sometimes with
> different per-item schemas for columns with different types.
> Furthermore, Lale includes functionality to automatically deduce JSON
> schemas from ARFF files, numpy ndarrays, pandas dataframes, etc.
> Then, Lale performs subschema checking: the schema of data passed
> needs to be a subschema of the schema of data expected, starting from
> the input and proceeding along all edges in a pipeline's dataflow graph.

---

### Official Review · Reviewer_D3TC · 2021-07-20

**Rating:** 3
**Confidence:** 4

**Summary:**

This paper proposes a combinators that can combine two functions without naming datasets. They implement these combinators as a part of sklearn pipeline---both prediction and training. Finally, it also adds a hyperparameter search using AutoML to optimize the models.

**Main Review:**

Strength

+ AutoML is an important application, and if successful, can greatly reduce data scientists' effort.
+ It looks like a significant engineering effort and mature tool. They support many backend for the optimizers.
+ Besides reporting testing accuracies for 14 OpenML classification tasks, they also reported  results of an user study demonstrating usefulness of the tool.

Weakness
- The high-level scientific contribution is missing. The paper provides too many low level details  without describing what are the main challenges to implement combinators.
- All the datasets used in this paper seems rather small, and it is not clear to whether the underlying framework can support diverse model architectures.
- It is not clear why the proposed method is contributing to AutoML. It seems like it is just combining existing AutoML approaches with the combinators.

**Time Spent Reviewing:**

2 hours

---

> ### Author Response · Authors · 2021-08-10
> **Author Response to Reviewer D3TC**
>
> > It is not clear why the proposed method is contributing to
> > AutoML. It seems like it is just combining existing AutoML
> > approaches with the combinators.
>
> The contribution of our paper is a library that offers a more
> convenient programming interface for AutoML.  This is well-aligned
> with the NeurIPS call for papers, which includes a category on
> "Infrastructure (e.g., datasets, competitions, implementations,
> libraries)". Also, NeurIPS has published papers on innovative
> programming interfaces for machine learning in the past.
>
> > The high-level scientific contribution is missing. The paper
> > provides too many low level details without describing what are the
> > main challenges to implement combinators.
>
> Lines 52-54 in the introduction crisply state the contributions of
> this paper. The main challenge is to support the modular
> combinator-based syntax by rewriting it to a form suitable for a given
> optimizer. This challenge is stated in Lines 215-217 of the paper.
>
> > All the datasets used in this paper seems rather small
>
> We chose OpenML datasets that allow for meaningful optimization (as
> opposed to just the initial few trials) with the default 1 hour
> setting of auto-sklearn. Our datasets are drawn from the AutoML
> Benchmark [1] and four of them were also used in the auto-sklearn
> evaluation. Also note that the goal of our evaluation is orthogonal to
> the sizes of the datasets.
>
> [1] Pieter Gijsbers, Erin LeDell, Janek Thomas, Sebastien Poirier,
> Bernd Bischl, and Joaquin Vanschoren, "An Open Source AutoML Benchmark".
> ICML Workshop on Automated Machine Learning (AutoML@ICML), 2019.
>
> > it is not clear to whether the underlying framework can support
> > diverse model architectures.
>
> Since Lale supports arbitrary nesting of simple core constructs, the
> diversity of supported architectures is one of its strengths. For
> instance, Table 2 presents results for four significantly different
> model architectures (Lale-Auto, Lale-TPOT, Lale-AD3M, Lale-ADB), and
> Section A.2 in the supplemental material shows details for them.
> We have also successfully used Lale on other data modalities, such as
> images, text, and time series. Furthermore, Lale also works with
> deep-learning operators, such as BERT embeddings.

---

### Decision · Program_Chairs · 2021-09-28

**Decision:**

Accept (Poster)

**Comment:**

Overall there is not enough support from reviewers for me to recommend acceptance.

Reviewers agreed on some real strengths in the paper, including (1) that is well-written and well-organized, and (2) that it tackles an important problem.

On #1:
* Reviewer G6BM: "I really enjoyed reading this paper! The paper is extremely well-written and well-organized"
* Reviewer cjD3: "The authors introduced their AutoML library in a sorted, logical way"

On #2:
* Reviewer G6BM: "The paper considers a very important problem and provides a very satisfactory solution both in technical and practical point of view"
* Reviewer D3TC: "AutoML is an important application, and if successful, can greatly reduce data scientists' effort"
* Reviewer H7JW: "AutoML is a useful tool [...] and efficient autoML system is an interesting research topic"

But the weight of opinion was that the paper (1) doesn't present a clear and significant scientific contribution, (2) shows limited empirical validation, and (3) doesn't concern a software package with large enough practical impact on the NeurIPS community.

On #1:
* Reviewer cjD3: "Furthermore, I don’t see a clear path for scientific future work building on top of Lale" and "I'm not convinced that the new operators can express substantially more than the formats of already existing AutoML tools [...] If the expressiveness of the format would be larger than prior work, I would have expected that we need also specialized optimizers for this [...] I have the impression that Lale could be a convenient package for new AutoML users, but this alone does not justify a NeurIPS paper"
* Reviewer H7JW: "be more clear about the scientific contribution from the very beginning of the paper, e.g., a more expressive formalization of the AutoML system."
* Reviewer D3TC: "The high-level scientific contribution is missing. The paper provides too many low level details without describing what are the main challenges to implement combinators." and "It is not clear why the proposed method is contributing to AutoML"

On #2:
* Reviewer cjD3: "the user studies only include 9 participants, making statements not very meaningful. Additionally, the survey might even be biased towards Lale"
* Reviewer H7JW: "organize the presentation of the empirical study following the general scientific principles; and to recruit more participants if possible so that one could draw some statistically significant conclusions."

On #3:
* Reviewer cjD3: "Overall, I’m not fully convinced that there will be many users of Lale at the end of the day." and "Lale is not even close to the level of pytorch. It has 225 stars on github and was forked 52 times. If that would be the level of impact we expect from a NeurIPS software paper, we will have thousands of papers of those each year."

Reviewer G6BM disagreed on #1 and #3, and felt the paper does clear the bar, but offered this follow-up commentary: "I suggest the authors address those concerns in the next version of the paper to make the paper stronger, e.g., by highlighting the first point more and providing concrete/quantitative evidence on how much useful the Lale library is (and will be) for the NeurIPS community."


The other reviewers offered several suggestions for the authors to improve the paper, in addition to those quoted above:
* Reviewer cjD3: "I liked Section 3 regarding the gradual automation which is a real problem for new (Auto)ML practitioners and I believe that there is a lot of untouched potential here. Unfortunately, this is not really the main focus of the paper."
* Reviewer cjD3: "Maybe JMLR MLOSS would be a better fit for this paper"


Overall, we weren't able to justify acceptance based either on the significance of the scientific contribution or on the practical impact of the software described here within the NeurIPS community. For that reason I recommend rejecting the paper, but I hope that the reviewers' feedback is helpful to the authors in improving it.

**Consistency Experiment:**

NeurIPS has a long history of experimentation. In 2014, NeurIPS ran an experiment in which 10% of submissions were reviewed by two independent committees to quantify the randomness in the review process. This year, we repeated a variant of this experiment to see how the quality of the review process has changed over time.  This paper was part of the experiment and was therefore assigned to two committees (consisting of reviewers, an Area Chair, and a Senior Area Chair) that reached independent decisions.  If both committees made the same recommendation, this recommendation was followed. If a single committee recommended acceptance, the paper was accepted (with the exception of a few cases in which the other committee identified what we considered a fatal flaw, e.g., an error in a key result).

This copy’s committee reached the following decision: **Reject**

The other committee assigned to the paper recommended **Accept (Poster)**.  You can find the other set of reviews, along with any follow up discussion with the authors here:
https://openreview.net/forum?id=wnAN2ZU7br